# Stability dependent increases in liquid water with droplet number in the Arctic

Rebecca J. Murray-Watson[1] and Edward Gryspeerdt[1]

[1]Space and Atmospheric Physics Group, Imperial College London, UK

**Correspondence:** Rebecca J. Murray-Watson (rebecca.murray-watson17@imperial.ac.uk)

**Abstract.** The effects of aerosols on cloud microphysical properties are a large source of uncertainty when assessing anthropogenic climate change. The aerosol-cloud relationship is particularly unclear in high-latitude polar regions due to a limited number of observations. Cloud liquid water path (LWP) is an important control on cloud radiative properties, particularly in the Arctic, where clouds play a central role in the surface energy budget. Therefore, understanding how aerosols may alter cloud LWP is important, especially as aerosol sources such as industry and shipping move further north in a warming Arctic.

Using satellite data, this work investigates the effects of aerosols on liquid Arctic clouds over open ocean by considering the relationship between cloud droplet number concentration ($N_d$) and LWP, an important component of the aerosol-LWP relationship. The LWP response to $N_d$ varies significantly across the region, with increases in LWP with $N_d$ observed at very high latitudes in multiple satellite datasets, with this positive signal observed most strongly during the summer months. This result is in contrast to the negative response typically seen in global satellite studies and previous work on Arctic clouds showing little LWP response to aerosols.

The lower tropospheric stability (LTS) was found to be an important control on the spatial variations in LWP response, strongly influencing the sign and magnitude of the $N_d$-LWP relationship, with increases in LWP in high stability environments. The influence of humidity varied depending on the stability, with little impact at low LTS but a strong influence at high. The mean $N_d$ state does not dominate the LWP response, despite the non-linearities in the relationship. As the $N_d$-LWP sensitivity changed from positive to negative when moving from high to low LTS environments, this work shows evidence of a temperature-dependent aerosol indirect effect. Additionally, the LWP-LTS relationship changes with $N_d$, generating an aerosol-dependent cloud feedback. As the LTS is projected to decrease and the boundary layer to become more polluted in a future Arctic, these results show that aerosol increases may produce lower cloud water paths. This shift to more unstable environments implies that LWP adjustments shift from enhancing the Twomey effect by 9% to offsetting it by around 50%, with this warming effect having potential consequences for sea ice extent.

## 1 Introduction

Aerosols can strongly influence the radiative properties of clouds through the modification of cloud microphysical properties. Some aerosols act as cloud condensation nuclei (CCN), and an increase in these aerosols leads to an increase in cloud droplet number concentration ($N_d$). For a constant cloud liquid water path (LWP), this leads to a decrease in cloud droplet radius,

which increases cloud albedo (Twomey, 1977). This would lead to a shortwave cooling effect at the top of atmosphere and at the surface. Smaller droplets may also have a smaller coalescence rates, and therefore delay the formation of precipitation (Albrecht, 1989). This leads to larger cloud LWP, which also increases cloud albedo. However, an increase in aerosol may also deplete LWP; smaller droplets cool and evaporate more quickly, generating turbulence and accelerating the entrainment of dry air into the cloud (Ackerman et al., 2004; Xue and Feingold, 2006; Williams and Igel, 2021). This promotes further cloud evaporation, which reduces the cooling effect of the cloud. Buffering mechanisms in the system make the influence of aerosols on cloud properties difficult to deduce (Stevens and Feingold, 2009); for example, aerosol-induced instability may also deepen the cloud layer, creating higher LWP clouds which in turn precipitate more, offsetting the LWP gains from the precipitation suppression mechanism.

The size and magnitude of the effects of aerosols on cloud LWP, and therefore net radiative effect, are uncertain. Modelling studies often find increases in LWP with aerosols (Quaas et al., 2008), whereas satellite-based satellite studies typically observe weak or negative responses (e.g., Michibata et al., 2016; Malavelle et al., 2017; Gryspeerdt et al., 2019). Meteorological conditions strongly influence the sign and magnitude of the relationship, with increases in LWP with aerosol loading typically observed in humid conditions (Coopman et al., 2016; Toll et al., 2019).

The relationship between aerosols and Arctic clouds is particularly unclear, in part due to difficulties in obtaining observations (Grosvenor and Wood, 2014). However, as industrialisation moves to higher latitudes, understanding how aerosols change cloud properties will become increasingly important (Schmale et al., 2018). This is particularly essential as low-level liquid-containing clouds play a central role in the Arctic energy budget, in which they often contribute to surface heating through their longwave warming effect (Curry and Ebert, 1992; Shupe and Intrieri, 2004). This contrasts with the rest of the globe, where the shortwave cooling effect dominates (L'Ecuyer et al., 2019). The difference in the Arctic is attributed to two key phenomena; polar night, during which the shortwave cooling effect is non-existent, and the presence of bright surfaces such as snow and ice. Overlying clouds cannot reflect significantly more radiation than these high-albedo surfaces, which again negates their cooling effects. However, Arctic clouds may have a cooling effect in the summer months, when sea ice retreats and there is ample solar radiation (Intrieri et al., 2002). The warming effect of clouds have been linked with sea ice loss (Kay and Gettelman, 2009; Huang et al., 2019) and melting of the Greenland ice sheet (Bennartz et al., 2013).

Previous in-situ and satellite studies have shown that Arctic clouds are more sensitive to anthropogenic aerosols than their low latitude counterparts (Garrett et al., 2004; Coopman et al., 2018). The resulting changes in their microphysical properties can substantially change the net radiative effect of the clouds (Lubin and Vogelmann, 2006; Zhao and Garrett, 2015), with the magnitude and sign of the effect also dependent on other factors including season and the albedo of the underlying surface. These modifications in cloud properties may have significant implications for the Arctic, which is undergoing rapid environmental change. The region is warming at an accelerated rate, at least twice the global average (Serreze and Barry, 2011), a phenomenon known as Arctic Amplification. Although primarily driven by an increase in greenhouse gas emissions, clouds play an uncertain role, with models predicting a wide range in the magnitude of their effect (Pithan and Mauritsen, 2014). Aerosol-induced changes to the cloud radiative effects may cause clouds to amplify or counteract this phenomenon (Schmale et al., 2021).

Identifying the role of aerosols on cloud properties is further complicated by the influence of confounding variables. For example, increases in satellite-retrieved aerosol optical depth (AOD) due to aerosol swelling in high humidity conditions may generate spurious correlations between aerosol and cloud properties (e.g., Quaas et al., 2010). Coopman et al. (2016) found that if meteorology is not accounted for, the magnitude of the cloud response to aerosol is larger by a factor of three than when the confounding influence of meteorology is reduced by using reanalysis data. To circumvent these issues, recent studies (Gryspeerdt et al., 2016) have used a mediating variable, such as $N_d$. $N_d$ is a good choice as its retrieval is not strongly affected by relative humidity and the impact of aerosols on LWP acts through changes to the cloud droplets. By considering $N_d$, the relationship between LWP and aerosols can be broken down into two parts; this can be represented using the sensitivity parameter (Feingold et al., 2001), which quantifies the relative change in LWP for a change in AOD (or $N_d$):

$$\frac{d \ln LWP}{d \ln AOD} = \frac{d \ln LWP}{d \ln Nd} \frac{d \ln Nd}{d \ln AOD} \tag{1}$$

The use of $N_d$ is particularly helpful in the Arctic; the persistently high cloud fraction (Shupe, 2011; Cesana et al., 2012) and high albedo surfaces means passive sensors (such as MODIS) can only obtain limited valid aerosol retrievals.

Although their high temporal resolution and large spatial coverage overcome the issues faced by in-situ measurements and field campaigns, few previous studies have used satellites to study Arctic aerosol-cloud interactions (Coopman et al., 2018; Zamora et al., 2018; Maahn et al., 2021). This work uses several years of satellite data from multiple instruments to investigate the relationship between LWP and $N_d$, using reanalysis data for investigate the influence of meteorology. The findings suggest that the lower tropospheric stability (LTS) is a dominant control in the $N_d$-LWP relationship, which may have significant implications in a warmer, ice-free Arctic.

## 2 Materials and Methods

Observational data used in this study are obtained from the Moderate Resolution Imaging Spectroradiometer (MODIS) on board NASA's Aqua satellite for the years 2010 to 2015, inclusive, using the cloud properties from the level 2 collection 6.1 dataset (MYD06_L2; Platnick et al., 2017). Only pixels above 60° latitude were included in this work. The data were regridded from their native 1 km by 1 km resolution to 25 km by 25 km and into the polar stereographic projection. The analysis is performed at an orbital level to avoid temporal averaging of the data. The data were filtered to include only single layer liquid clouds using the 'Cloud_Phase_Infrared', 'Cloud_Phase_Optical_Properties' and the 'Cloud_Multi_Layer_Flag'. Liquid-topped mixed-phase clouds are common in the Arctic, and the MODIS cloud phase algorithm may incorrectly classify these clouds as purely liquid clouds. As such, only pixels with cloud top temperatures above 268 K were included in this study, as in situ measurements show these clouds have a liquid water fraction of upwards of 95% (de Boer et al., 2009). MODIS has been shown to underestimate cloud top temperatures in the Arctic (Tietze et al., 2011), meaning that this filtering step is likely removing too many clouds. However, as the inclusion of mixed-phase clouds would introduce significant uncertainties (Khanal and Wang, 2018), this conservative estimate of the cloud top temperature is used.

The cloud liquid water path was estimated according to Equation 2:

$$LWP = \frac{5}{9}\rho_w \tau_c r_e \tag{2}$$

in which $\rho$ is the density of water and $\tau_c$ and $r_e$ are the cloud optical thickness and the cloud droplet effective radius, both acquired from MODIS. Equation 2 assumes adiabatic conditions, such as the cloud interacting with the environment through precipitation or entrainment (Brenguier et al., 2000; Wood and Hartmann, 2006).

For comparison with the MODIS data, LWP data was also obtained from version 2 of the Advanced Microwave Scanning Radiometer for EOS (AMSR-E) ocean product, which is also aboard Aqua (Wentz and Meissner, 2004). Data from 2010 and 2011 were included in the analysis. The data were regridded from their native 12 km by 12 km grid to the same 25 km by 25 km polar stereographic grid as the MODIS data. The in-cloud LWP is calculated by dividing the AMSR-E LWP by the MODIS liquid cloud fraction.

The cloud droplet number concentration ($N_d$) is estimated from MODIS data using Equation 3:

$$N_d = \frac{1}{2\pi k}\sqrt{\frac{5}{Q\rho_w}}(f_{ad}c_w)^{\frac{1}{2}}\tau_c^{\frac{1}{2}}r_e^{-\frac{2}{5}} \tag{3}$$

in which $k$ is associated with the droplet spectrum width and is assumed to have a constant value of 0.8 (Painemal and Zuidema, 2011; Grosvenor and Wood, 2014). Q is the scattering coefficient as is approximately equal to 2 (Bennartz, 2007). The condensation rate depends on temperature and weakly on pressure, although the pressure dependence is weak. Therefore, the condensation rate calculated using the linear relationship derived in (Gryspeerdt et al., 2016), using the MODIS cloud top temperature (following Grosvenor and Wood, 2014). The subadiabatic factor ($f_{ad}$) represents the degree to which the cloud departs from the adiabatic profile. While previous work has found that marine stratiform clouds are generally found to be close to adiabatic (Zuidema et al., 2005), other studies have shown $f_{ad}$ can vary widely (e.g. Merk et al., 2016). This study assumes $f_{ad}$ of 0.7, following Painemal and Zuidema (2011).

Several unique aspects of the Arctic environment, including high solar zenith angles, high albedo surfaces such as sea ice and snow and persistent darkness during polar night, can create challenges for passive satellite sensors to obtain accurate retrievals of clouds. To detect clouds, MODIS relies on contrast between the clouds and the underlying surface in thermal and visible channels. During polar night, MODIS is unable to use visible bands to detect clouds, and therefore must solely rely on infrared channels. Chan and Comiso (2013) found that due to insufficient thermal contrast between clouds and the surface, MODIS frequently failed to discriminate between them during night time retrievals. Additionally, sea ice or snow-covered surfaces were frequently flagged as cloud due to similar temperatures or reflectivity, with 30.9% disagreement between MODIS and CALIOP, an active sensor, in detection of cloudy scenes over these surfaces. Due to greater thermal and visible contrast between clouds and the surface, MODIS performed better over ocean, with only 3.7% disagreement in cloud detection between MODIS and CALIOP.

High albedo surfaces also pose difficulties for retrievals of cloud microphysical properties. Dong et al. (2016) showed the difference between satellite-retrieved $\tau_c$ and that measured by in-situ instruments and Utqiaġvik (formerly Barrow) was higher

in snow-covered than snow-free regions (which included ocean) and increased at higher surface albedo due to lack of contrast
between cloud and surface.

Large solar zenith angles, which occur frequently at high latitudes, introduce additional biases in cloud property retrievals. Grosvenor and Wood (2014) found that $\tau_c$ increases substantially with SZA above 65-70°. Retrieved $r_e$ decreases at higher SZA, leading to an overestimate of $N_d$ by 50-70%, depending on the MODIS band used and the sensor viewing angle. Over snow-free surfaces, Dong et al. (2016) saw also saw an increases in retrieved $\tau_c$ and $r_e$ relative to in situ measurements at very high SZA, generating overestimate in LWP relative to the ground measurements of about 20 g m$^{-2}$ at SZA greater than 72°.

Careful filtering of the L2 pixels to remove cases in which the data are known to be highly uncertain can prevent the introduction of biases into the results in spite of these difficult conditions. As such, to limit these uncertainties, the pixels were filtered only to include those with a 5 km cloud fraction of above 0.9 to limit uncertainties associated with retrievals at the cloud edge. Pixels with a heterogeneity index ('Cloud_Mask_SPI') above 30 were removed, as inhomogeneous clouds are known to introduce retrieval biases (Zhang and Platnick, 2011). The solar zenith angle and the sensor viewing angle were limited to 65° and 50°, following Grosvenor and Wood (2014). Requiring low solar zenith angles means that only sunlit pixels are included in this analysis.

Due to uncertainties associated with retrievals of cloud properties over snow and ice covered surfaces by passive sensors, only ocean pixels were considered. The sea ice pixels were removed using daily sea ice cover data from Nimbus-7 SMMR and DMSP SSM/I-SSMIS Passive Microwave Data, Version 1 dataset (Cavalieri et al., 1996), also gridded at a 25 km by 25 km resolution. Open ocean pixels adjacent to sea ice-containing pixels were also removed from this analysis to minimise the impacts of undetected sea ice.

The $N_d$ retrievals were further limited to include only pixels with an $r_e$ greater than 4 $\mu$m and a $\tau_c$ greater than 4. This is due to uncertainties associated with retrievals of smaller values (Quaas et al., 2006; Sourdeval et al., 2016). This stringent filtering is not applied to the LWP retrievals as $N_d$ is more sensitive to inaccuracies in these values and would introduce a high bias in the MODIS LWP against the AMSR-E LWP (Gryspeerdt et al., 2019). A summary of the filtering methods has been provided in Figure S1.

Meteorological reanalysis data was obtained from the ERA5 dataset (Hersbach et al., 2020), produced by the European Centre for Medium-Range Weather Forecasts. The data at the time step which is closest to that of the time of the satellite overpass is considered to be temporally coincident for this study. The data were gridded from their original 0.25° by 0.25° grid onto the same 25 km by 25 km grid as the MODIS data. The effects of free tropospheric moisture and lower tropospheric stability (LTS) were studied as previous works have shown these variables have a strong influence the $N_d$-LWP relationship (Chen et al., 2014; Michibata et al., 2016; Coopman et al., 2016). The specific humidity at 750 hPa ($q_{750}$) was chosen as a measure of the humidity of the free troposphere. The LTS, which is a measure of the static stability of the atmosphere, was calculated as the potential temperature difference between 700 hPa and 1000 hPa (Klein and Hartmann, 1993).

Additionally, the marine cold air outbreak index (MCAO; Kolstad and Bracegirdle, 2008) is important to cloud formation and behaviour at high latitudes (McCoy et al., 2017). Much like LTS, it is a metric of the stability of the boundary layer, although it is calculated as the difference between the potential temperature at 800 hPa and the sea surface temperature (Fletcher et al.,

2016), with positive values indicating higher instability. This metric is particularly suitable for the Arctic as it highlights the difference in temperature of the relatively warm ocean with the cool overlying air masses. The ocean-air temperature gradient is an important driver of boundary layer instability in the Arctic (Kay and Gettelman, 2009).

## 3 Results

### 3.1 The regional and seasonal $N_d$-LWP relationship

Figure 1 (a) shows the annual mean linear sensitivity of the MODIS LWP to the $N_d$. There is a clear negative-to-positive gradient in the sensitivity, with increases in LWP with $N_d$ typically occurring at higher latitudes. The seasonal cycle of the MODIS sensitivity is shown in Figures 1 (c), (d) and (e). Although spring and autumn have regions of positive sensitivity, the summer months most strongly contribute to the signal observed when considering the all-season response, as they have the most data. Figure 1 (b) shows the linear sensitivity between MODIS $N_d$ and AMSR-E LWP for all seasons. The positive relationship and spatial pattern is similar to the MODIS data, with the AMSR-E LWP show a stronger positive response to aerosols. This may be due to a potential negative bias in the MODIS data due to $r_e$ retrieval errors (Gryspeerdt et al., 2019). Due to the absence of incoming solar radiation during polar nights, there is a lack of data for the winter season.

The decrease in LWP with $N_d$ at lower latitudes has previously been observed in liquid-phase clouds over subtropical oceans (e.g. Michibata et al., 2016; Gryspeerdt et al., 2019) and is consistent with the mechanism of aerosol-enhanced entrainment mixing (Ackerman et al., 2004). Positive sensitivities have previously been observed in high relative humidity conditions (Chen et al., 2014; Toll et al., 2017). Increases in LWP also occur in very clean conditions due to precipitation suppression in this low $N_d$ regime (Gryspeerdt et al., 2019).

There are several possible explanations for the spatial heterogeneity in the LWP response. One potential cause could be due to air masses moving off of the ice edge; cold air moving over the relatively warm open-ocean regions has previously been associated with Arctic cloud formation (Pithan et al., 2018). However, the lack of a strong positive response around the ice edge during the spring months (Figure 1 (c)) suggests that the exposure of these air masses to new aerosol and moisture sources as they transition from the ice pack suggests this is not significant to the $N_d$-LWP relationship. The remaining potential drivers include the cloud mean state and meteorological conditions, which are investigated further in this work.

### 3.2 The role of meteorology and cloud mean state

To investigate the drivers of the different LWP responses, Figure 2 shows the distributions of cloud microphysical properties and meteorological variables for grid boxes with positive and negative sensitivity. There is no evidence for a statistically significant difference between the distributions of the LWP, specific humidity, $N_d$ vertical velocity or surface wind speed for two regions (Mann Whintey U test, p > 0.05; Mann and Whitney, 1947) . However, the distributions for the LTS, the surface temperature and the MCAO index are significantly different. In particular, in the region of positive sensitivity, the LTS tends towards higher values, whereas the surface temperature and the MCAO index are both lower. This influence of stability on the

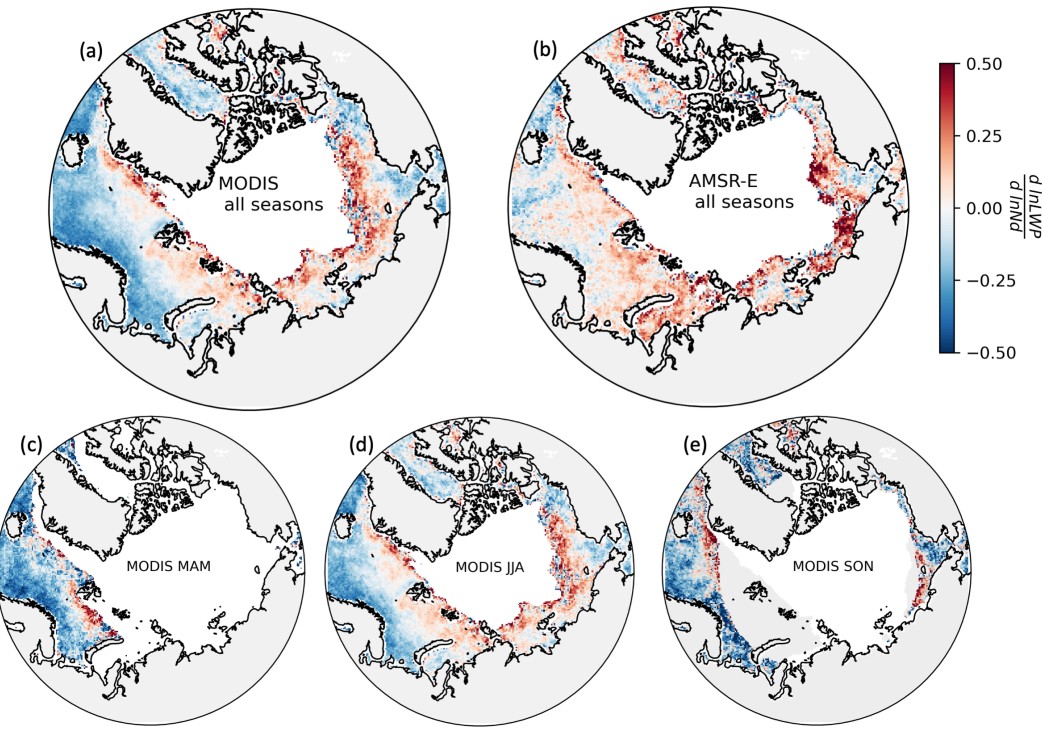

**Figure 1.** The sensitivity of the cloud liquid water path to $N_d$ for (a) MODIS all seasons, (b) AMSR-E all seasons, (c) MODIS March, April and May, (d) MODIS June, July and August and (e) MODIS September, October and November. White indicates data omitted due to the presence of sea ice while grey shows either missing data due to polar night or the presence of land masses.

LWP sensitivity is consistent with previous studies (Chen et al., 2014); higher stability conditions inhibit mixing between the cloud layer with the dry above-cloud layer, and prevents the depletion of LWP due to the evaporation-entrainment mechanism.

The $r^2$ values of the correlation between the sensitivity and mean of meteorological variables for each pixel over the six years of data was calculated. The $r^2$ is higher for the LTS (0.39) than for the MCAO index (0.26) and surface temperature (0.32); the associated scatter plots are shown in Figure S5. This indicates that the LTS explains a greater fraction of the variance in the

sensitivity than the other meteorological variables. Due to its better performance as an explanatory variable, LTS is used in the remainder of this study as a proxy for the importance of stability and surface forcing on cloud formation.

This association between positive sensitivities and higher LTS values explains why the strongest positive signal was observed during JJA in Figure 1. During the summer, the ocean temperature is constrained by melting sea ice, while the atmosphere is warmed by an increase in solar radiation, creating a weak ocean-air temperature gradient (Kay and Gettelman, 2009; Persson,

2012; Morrison et al., 2018). This results in warm air residing above a relatively cold ocean, producing high LTS conditions. Conversely, in spring and autumn, lower stability conditions are more common. Due to its large thermal heat capacity, the ocean

remains warm relative to the atmosphere in seasons with low insolation, generating a larger ocean-air temperature gradient and unstable conditions (Kay and Gettelman, 2009; Morrison et al., 2018; Huang et al., 2019).

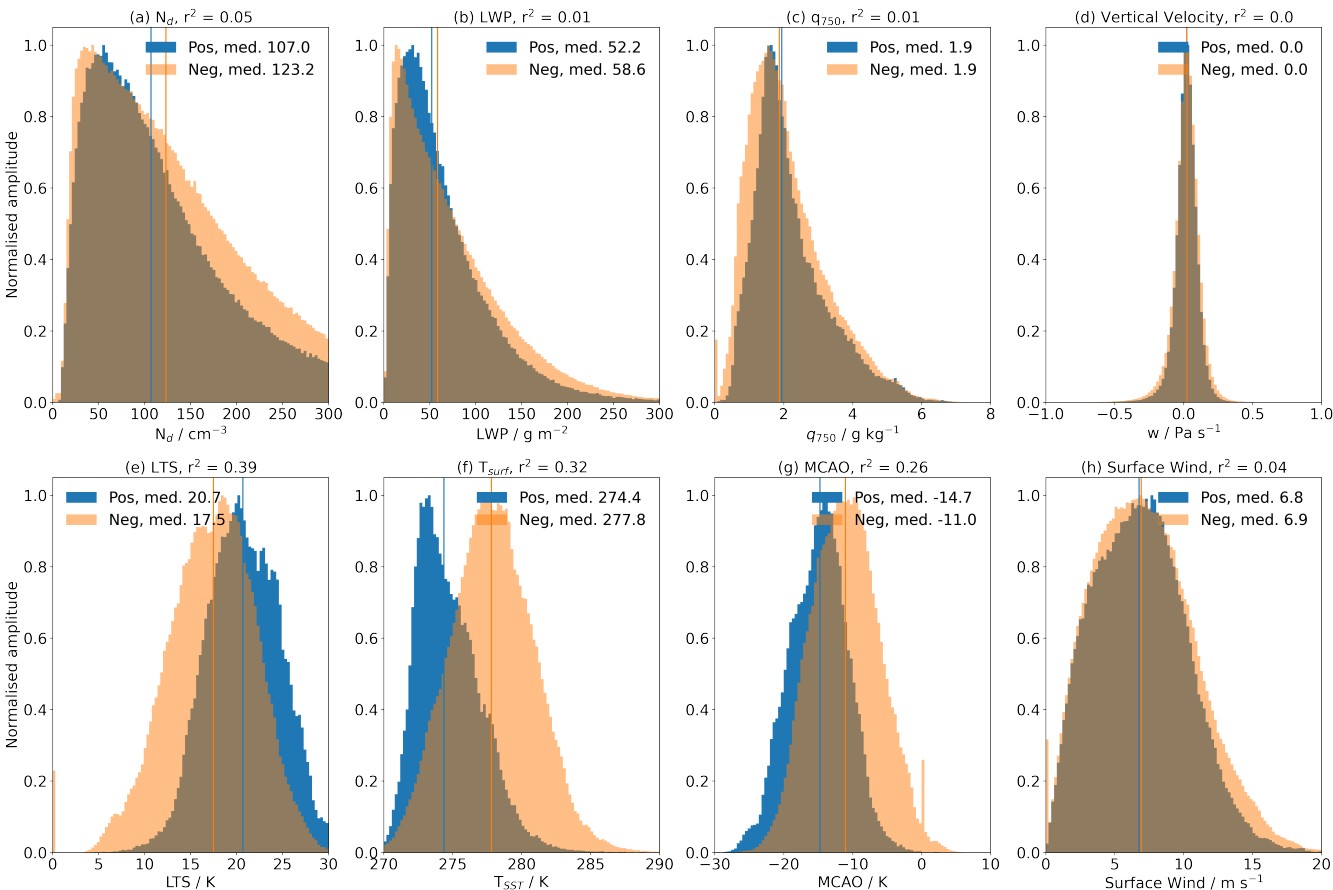

**Figure 2.** Normalised histograms showing the distributions of (a) $N_d$, (b) LWP, (c) $q_{750}$, (d) vertical velocity, (e) LTS, (f) $T_{surf}$, (g) MCAO and (g) surface wind for the regions of positive and negative sensitivity shown in Figure 1 (a). The blue and orange vertical lines show the medians for the positive and negative regions, respectively. The $r^2$ values relate to the correlation between the sensitivity and mean of meteorological variables for each pixel over the six years of data was calculated.

From Figure 2, it appears that the specific humidity does not strongly drive the LWP response. However, previous work has shown that specific humidity often strongly influences the $N_d$-LWP relationship, with more weakly negative responses under higher humidity conditions (Chen et al., 2014; Toll et al., 2019) due to a suppressed evaporation-entrainment mechanism. To investigate this further, the data were partitioned into bins of specific humidity and LTS (Figure 3). The response to changes in specific humidity is weak, with strongly negative sensitivities evident under both humid and dry conditions. The response to variations in LTS is greater, with a strong negative response at low LTS turning into a positive sensitivity in higher stability

conditions. The strong dependence on stability supports the hypothesis that the LTS is the predominant control behind the differences between the regions of positive and negative sensitivity (Figure 2).

Although the overall response to humidity is weak, Figure 3 shows that its role is dependent upon the LTS conditions. The influence of humidity on the $N_d$-LWP relationship is small at low LTS; the sensitivity is consistently strongly negative across the humidity range. However, it becomes important at high LTS, where the response changes from negative to positive as humidity

increases. This relationship holds when using specific humidity from different vertical levels (Figure S3). These results are similar to previous work on Arctic clouds; in high stability environments, (Coopman et al., 2016) found an increasingly positive response with $q_{750}$. Additionally, the sensitivity to aerosol increased with LTS when $q_{750}$ was constrained between 2.0 and 4.0 g kg$^{-1}$.

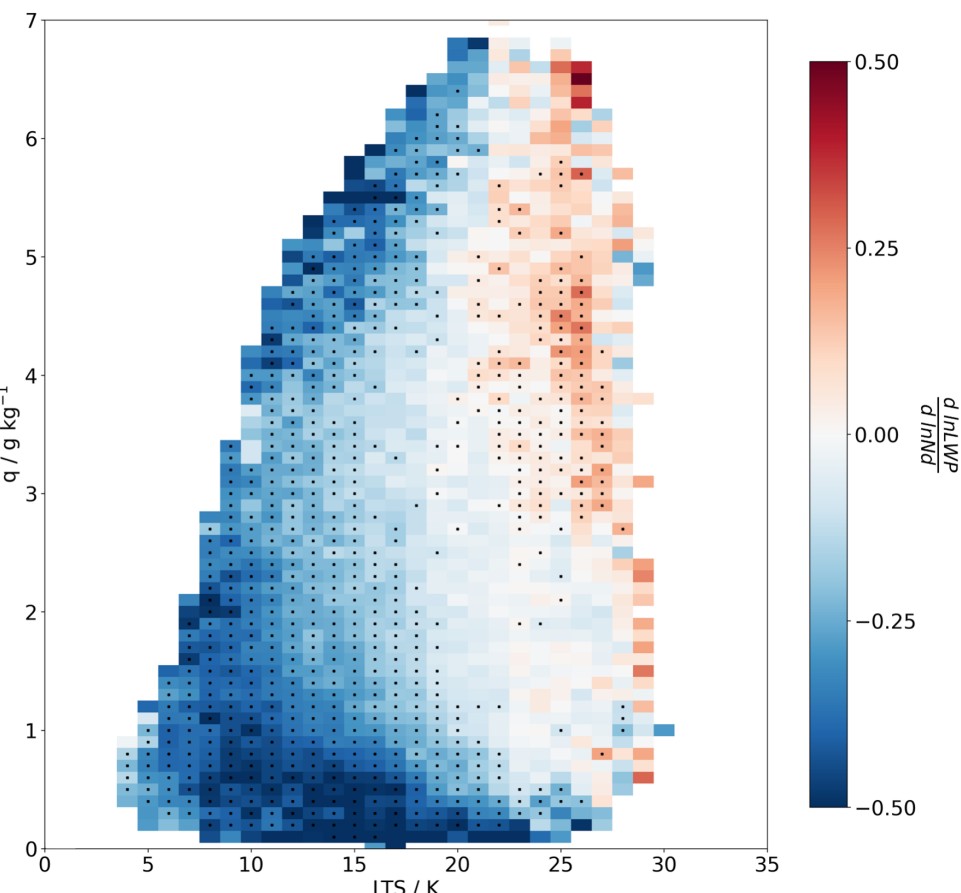

**Figure 3.** The linear $N_d$-LWP sensitivity plotted as a function of LTS and cloud-top humidity ($q_{750}$). Only bins which included over 100 valid retrievals were included in the analysis. The black dots indicate points for which the correlation is significant at a 95% confidence level.

The results presented so far have assumed a linear sensitivity of LWP to $N_d$; however, assuming linearity means important
characteristics of the relationship are not considered. For example, the response of LWP can be non-linear, with a dependence
on the initial cloud state (Gryspeerdt et al., 2019). Additionally, use of the linear sensitivity parameter does not consider the
absolute values of the LWP and how these change in different meteorological regimes. To investigate these characteristics,
joint probability histograms were generated by creating a 2D histogram of LWP and $N_d$ and then normalising each column
by the total $N_d$, such that each pixel in that column represented P(LWP|$N_d$), or the probability of observing a particular LWP
given a particular $N_d$. These diagrams also allow for exploration of where along the $N_d$ spectrum the meteorological conditions
considered in Figure 3 become important to the $N_d$-LWP relationship.

Figure 4 shows the $N_d$-LWP joint probability histogram for four different environmental regimes, partitioned into high and
low LTS and specific humidity bins. The differences between the normalised histograms are shown at the end of each row and
column, and the $N_d$ distributions for the regimes are shown beneath each histogram. The blue lines on the difference plots
shows clouds with droplet effective radius of 15 $\mu$m; clouds to the left of this line are assumed to be precipitating. The black
lines on the joint probability histograms are the mean LWP for each $N_d$ bin, with the shading showing the 95% confidence
interval. These mean LWP lines are expanded in Figure 5 for clarity. Figure 5 also displays the seasonal breakdown of this
relationship, showing that the meteorological dependence of the $N_d$-LWP is robust across seasons.

In high LTS conditions, Figures 4 and 5 (a) and (b) show evidence of three different responses of LWP to $N_d$, which are
related to the precipitation behaviour as a function of $r_e$ and the pixel aggregation scale. First, in clean states (low $N_d$; up to
20 cm$^{-3}$), there are strong increases in LWP with $N_d$, as has been observed in previous work (Gryspeerdt et al., 2019); this
is consistent with precipitation suppression. There is little difference in the high and low $q_{750}$ environments in these low $N_d$
conditions (Figure 4 (c)), suggesting that the precipitation suppression mechanism is not strongly reliant on specific humidity.
The $N_d$ distributions below the histograms show that these very low $N_d$ conditions are relatively rare in the Arctic and therefore
have little bearing on the linear sensitivity.

In the second regime, between 50 cm$^{-3}$ and 150 - 200 cm$^{-3}$, LWP still increases with increasing $N_d$, but the strength
of the increase is not as large as for the clean cases (Fig 5 (a) and (b)). This is likely due to the precipitation suppression and
evaporation-entrainment mechanisms competing to modulate LWP. Figure 6 shows that at these droplet number concentrations,
there is still a significant proportion of the droplets which are sufficiently large ($r_e > 15 \mu$ m) for the precipitation suppression to
augment LWP. However, most droplets are smaller than 15 $\mu$ m, so the precipitation suppression mechanism is weaker than at
lower $N_d$. In high LTS conditions, the evaporation-entrainment mechanism is suppressed enough that $N_d$ increases up to 150 -
200 cm$^{-3}$ can still lead to increases in LWP. The similarity between the high and low humidity cases for this $N_d$ range in Figure
4 (c) also suggests that this $N_d$-LWP relationship in this regime is not dominated by the evaporation-entrainment mechanism.
From the $N_d$ distributions, the majority of the data is in this regime and therefore it strongly influences the $N_d$-LWP sensitivity.

Finally, the LWP decreases with further increases in $N_d$ in heavily polluted (above 200 cm$^{-3}$) environments. At these $N_d$,
droplets are rarely larger than 15 $\mu m$ (Figure 6), so the precipitation suppression mechanism can't generate further increases
in cloud water. Therefore, the evaporation-entrainment mechanism becomes the dominant control and begins to reduce LWP.
Figure 4 (c) and Figure 5 (b) show that the LWP decreases are stronger in low humidity environments due to the enhanced

evaporation of cloud droplets under these conditions. At high $q_{750}$, free tropospheric moisture can buffer against cloud water loss.

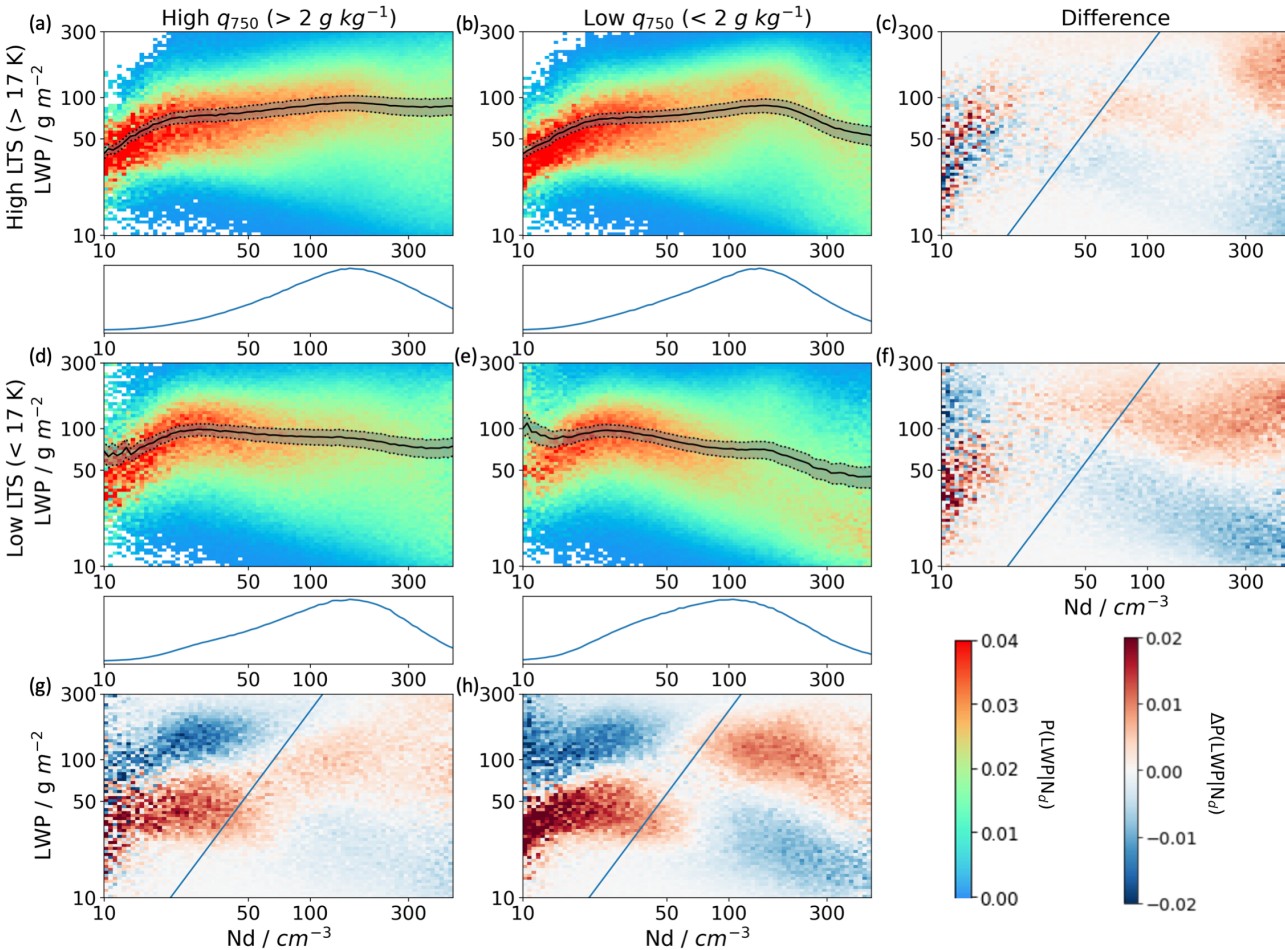

**Figure 4.** Joint probability histograms for the $N_d$-LWP divided into four meteorological regimes based on LTS and q. The difference plots are shown at the end of each row and column, with red over blue indicating higher LWP at higher humidity/LTS. The black lines and grey shading on the joint probability histograms represent the mean LWP value for each $N_d$ bin and the 95 % confidence interval, respectively. The blue lines on the difference plots indicate clouds with effective radius of 15 $\mu$ m, so clouds to the right of the line are expected to be non-precipitating.

Similarly to the stable regimes, the precipitation suppression mechanism is again evident for $N_d$ up to 20 - 30 cm$^{-3}$ in low LTS environments. After this, the LWP decreases with increasing $N_d$, with stronger decreases seen in low $q_{750}$ conditions (Figure 5 (c) and (d)). This is consistent with the evaporation-entrainment mechanism. Figure 4 (f) shows that the difference between the high and low humidity regimes manifests at a lower droplet number than in the high LTS regimes; in Figure 4 (c), clouds in high $q_{750}$ environments do not have significantly greater LWP than clouds in low $q_{750}$ environments until $N_d$

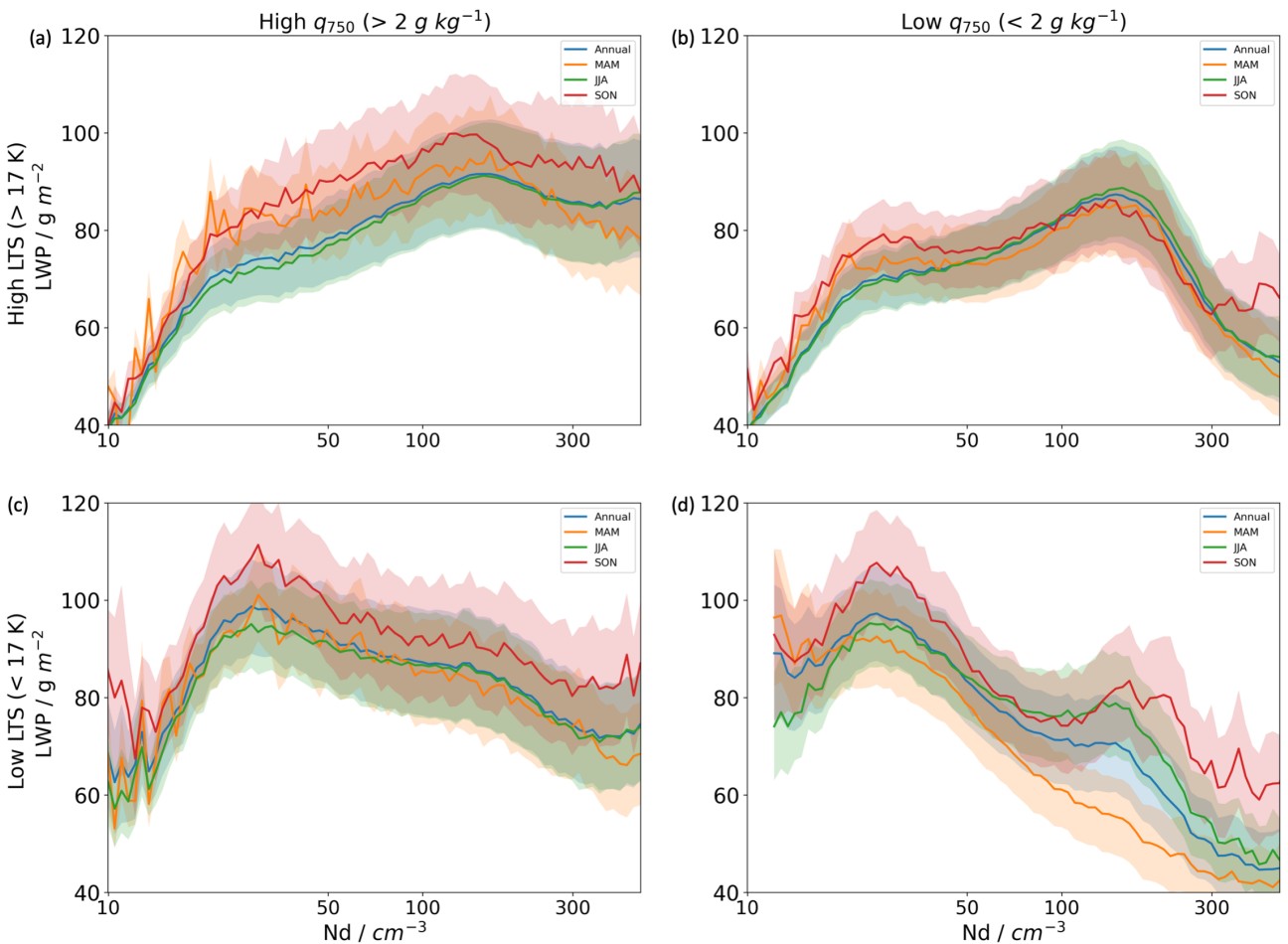

**Figure 5.** The mean LWP for each N$_d$ bin and the 95 % confidence interval for different meteorological regimes. The blue line is for all seasons and equivalent to the black lines in Figure 4, and the orange, green and red lines are for MAM, JJA and SON respectively.

reaches above 250 cm$^{-3}$ (indicated by the darker red over blue region Figure 4 (c)). However, in Figure 4 (f), these larger LWP values in moister conditions occur at earlier at around 40-50 cm$^{-3}$, with the disparity growing stronger with higher N$_d$. This importance of humidity at lower N$_d$ at low LTS may be due to increased turbulent mixing with the above-cloud layer. This would enhance the rate of droplet evaporation, thereby increasing the dependence of LWP on q$_{750}$.

When comparing across stability regimes, Figures 4 (g) and (h) show that when the N$_d$ is below 50 cm$^{-3}$, low LTS conditions support clouds with a higher LWP. This may be due to a deepening of the boundary layer under unstable conditions, allowing clouds to grow deep enough to precipitate, such that an increase in aerosol allows for more LWP to be retained in the cloud through the precipitation suppression mechanism. Additionally, low LTS facilitates the vertical transport of moisture, thereby promoting cloud formation (Kay et al., 2016). This behaviour is similar in both high and low q$_{750}$ environments, suggesting

that humidity plays a smaller role at low N$_d$.

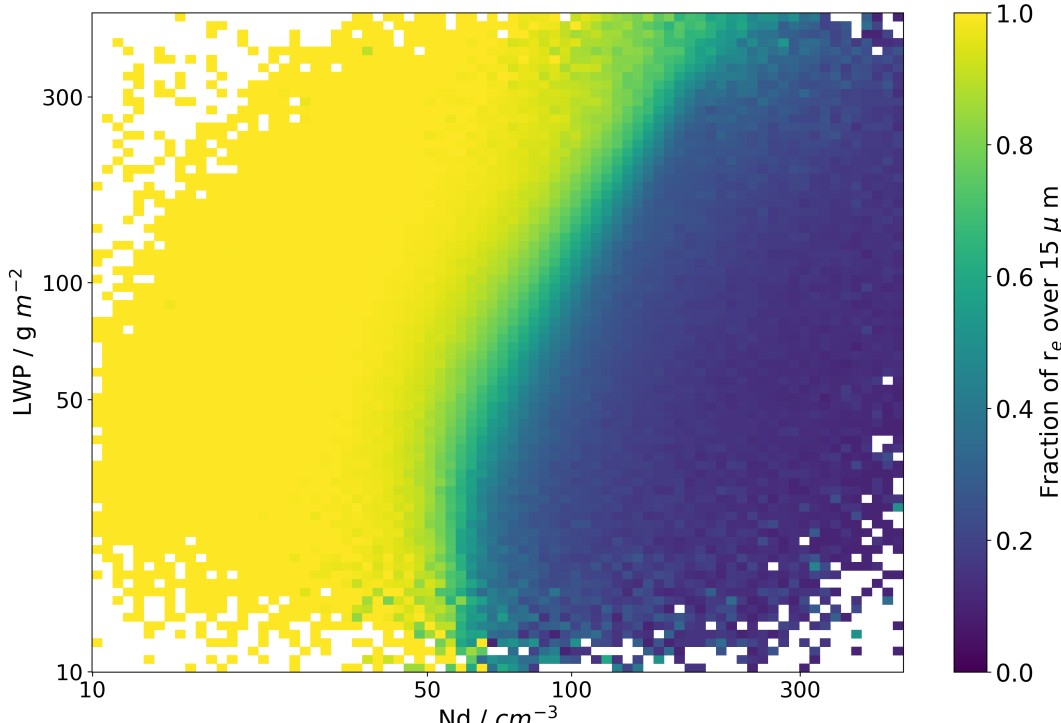

**Figure 6.** Fraction of 1km pixels have an $r_e$ >15 $\mu$m for a given 25km by 25 km grid, as a function of LWP and $N_d$. Droplets with effective radii above 15 $\mu$m are assumed to be precipitating. The threshold of 15 $\mu$m for collision-coalescence is taken following Rosenfeld and Gutman (1994).

However, as $N_d$ increases above 50-100 cm$^{-3}$, Figures 4 (g) and (h) show that higher LTS conditions support higher LWP clouds than low LTS conditions. Stronger mixing with subsaturated air in low LTS environments may enhance droplet evaporation relative to stable environments. Dryer cloud tops would strengthen this effect, resulting in the disparity in cloud LWP for high and low LTS being stronger in low q$_{750}$ environments (Figure 4 (h)).

## 3.3 Radiative impacts

The potential radiative impact of the LWP adjustments relative to the albedo enhancements from more numerous, smaller droplets (the Twomey effect) can be estimated from :

$$\frac{d \ln \alpha}{d \ln Nd} = \frac{1-\alpha}{3}(1 + \frac{5}{2}\frac{d \ln LWP}{d \ln Nd}) \tag{4}$$

In which $\alpha$ is the cloud albedo (following Platnick and Twomey, 1994). An estimate of the $N_d$-LWP sensitivity can be generated by using the present-day $N_d$ distribution and the joint probability histogram to generate a LWP distribution. The

present day $N_d$ distribution is then adjusted such that the mean increases by 10%, to represent a hypothetical increase in aerosol concentration in a future Arctic. A new LWP distribution is calculated using this adjusted $N_d$ distribution and the joint probability histograms. The change in the mean LWP and mean $N_d$ between the two sets distributions is calculated to estimate the sensitivity. Using this method, in high LTS conditions (ignoring the effects of humidity), LWP adjustments imply a 9% enhancement of the Twomey effect (slope of +0.03), and under low LTS conditions, an offset of 51% (slope of -0.17). The sign of the sensitivity is insensitive to the size of the perturbation for moderately sized adjustments to the $N_d$ distribution.

## 4    Discussion

In this work, we have investigated the factors which influence the $N_d$-LWP relationship in Arctic clouds. We have found that LTS is a dominant control on the LWP response, with increases in LWP possible in high stability conditions. Specific humidity only appears to influence the relationship in polluted or high LTS conditions, while the $N_d$ state exerts little control on the LWP response. However, despite careful filtering to remove instances in which the data are prone to errors, some uncertainties in the results remain.

For example, although the pixels have been filtered by cloud top temperature, the misclassification of mixed-phase clouds as liquid may influence the results. However, Khanal and Wang (2018) showed that the error associated with this misclassification is small when compared those generated by high solar zenith angles experienced in the Arctic, which in turn has been addressed in this work by the omission of high-angle pixels.

As LWP and $N_d$ are both calculated using the $\tau_c$ and $r_e$ retrievals, errors in these properties may result in significant correlated errors in the LWP and $N_d$ MODIS retrievals. Gryspeerdt et al. (2019) show that errors in $\tau_c$ would generate a positive bias, whereas errors in $r_e$ would create negative bias. However, when this analysis was repeated using the AMSR-E LWP (Figures 1 (b), S6 and S7), the LWP response to $N_d$ across the different meteorological regimes was consistent with the results using the MODIS LWP. AMSR-E is an independent dataset and therefore not affected by the correlated errors in $\tau_c$ and $r_e$. Therefore, while correlated errors may affect the LWP-$N_d$ relationship, they do not dominate the results and the observed relationships are not just retrieval artefacts.

Despite ERA5 performing better than other reanalysis datasets when compared to in-situ observations of temperature and humidity (Graham et al., 2019), the meteorological conditions in the Arctic are still poorly constrained. Therefore, use of the reanalysis data may introduce additional uncertainties into the results. However, Renfrew et al. (2021) found that ERA5 compared well to in-situ observations of ice-free regions in the Arctic, so these uncertainties are unlikely to strongly impact the findings of this work.

As the Arctic warms, the LTS is projected to decrease (Boeke et al., 2021). Figure 4 shows that in lower LTS environments, LWP typically decreases with $N_d$, which weakens shortwave cooling effect of the clouds on the surface. Assuming moderate increases in $N_d$, the LWP adjustments shift from amplifying the Twomey effect by 9% in a stable environment to a 51% reduction in an unstable one. This change in the sign of the $N_d$-LWP relationship with warming amounts to a temperature-

dependent indirect effect; in a warmer Arctic with a more unstable boundary layer, LWP adjustments may shift from enhancing the cooling effect to offsetting it.

Equally, these results also demonstrate an aerosol-dependent cloud feedback, as the LWP response to changes in LTS is different in clean and polluted environments. In more polluted environments, LWP decreases more strongly at low LTS. This weakens the negative cloud feedback, shifting to a positive effect. This influence of aerosols on the cloud feedback is key as industrialisation and the creation of new trans-Arctic shipping lanes are projected to be developed as the Arctic heats and sea ice retreats, introducing a large new source of anthropogenic aerosols (Peters et al., 2011; Schmale et al., 2018).

In stable conditions, the $N_d$-LWP relationship is positive, with decreases in LWP only seen in heavily polluted environments (Figure 4 (a) and (b)). Additionally, in relatively clean conditions, clouds in low LTS environments have higher LWP than clouds in stable environments (Figure 4 (g) and (h)). Therefore, individually, the increases in aerosol or decreases in LTS projected for the Arctic may act to strengthen the cloud shortwave effect. However, working together in a warmer Arctic they may produce clouds with lower water paths, leading to a weaker negative cloud feedback in a more polluted environment.

The results presented here are only for liquid clouds over ocean; more work is required using different datasets to see if they hold in ice-covered regions or for mixed-phase clouds. Nevertheless, these findings that the aerosol-cloud interactions change with warming and that the LWP-LTS relationship depends on the aerosol loading may have significant implications for the surface energy budget in a rapidly changing Arctic. For example, potential decrease in cloud LWP suggested by this study could have significant consequences for Arctic sea ice extent. Thinner clouds have a lower albedo, and therefore a lower shortwave cooling effect at the surface. In non-summer months, when surface albedo is low due to the presence of open ocean, this leads to an increase in solar radiation being absorbed by the surface. Previous work has found a negative correlation between the amount of radiation absorbed by the surface in summer, which is in part controlled by cloud LWP, and sea ice extent later in the year (Choi et al., 2014; Huang et al., 2017). The effects of these LWP changes to the longwave effect, which dominates in non-summer months, is expected to be weaker as changes in longwave downwelling radiation are more strongly controlled by cloud fraction as Arctic clouds in non-summer months typically have LWP greater than 30 g m$^{-2}$ and therefore act as black-body radiators (Shupe and Intrieri, 2004; Huang et al., 2017, 2019).

## 5 Conclusions

Previous studies have found a strong sensitivity of Arctic cloud properties and aerosols (Garrett et al., 2004; Coopman et al., 2018). However, these works were either of a limited spatial extent or considered the average response of cloud properties across the Arctic, and therefore did not observe the spatial heterogeneity in cloud response. This work considered the regional variation in the LWP response to N$_d$ in liquid clouds, documenting a positive sensitivity at higher latitudes. Positive relationships have previously been observed under some conditions, but not at the strength found in this work (Han et al., 2002; Chen et al., 2014; Toll et al., 2019; Gryspeerdt et al., 2019). However, the response is typically negative across the globe. The signal was most strongly observed during the summer months (Figure 1). Comparison of cloud and meteorological properties of the

regions displaying positive and negative sensitivity indicates that stability, in particular LTS, is a significant driving force for the difference in behaviour (Figure 2).

There is only a weak response to cloud-top specific humidity, but the variation with LTS was much greater (Figure 3). Under moist, stable conditions, the LWP increases with $N_d$, as seen with subtropical clouds (e.g. Chen et al., 2014). Even when considering cases with lower humidity, increases in LWP with $N_d$ are supported up until high $N_d$, at which point the humidity is insufficient to offset the moisture lost to droplet evaporation (Figure 4). The frequency of these high LTS conditions at high latitudes during the summer months explains the seasonal pattern in the sensitivity in Figure 1.

This work found evidence for a temperature-dependent aerosol indirect effect through the change in sign of the $N_d$-LWP relationship with LTS and an aerosol-dependent cloud feedback as the LWP response to LTS, and therefore surface temperature, changes with $N_d$. Unstable conditions generate higher LWP values than stable conditions for low $N_d$, potentially due to precipitation suppression (Figure 4). Therefore, in a future, lower-LTS environment, clouds have a stronger shortwave cooling effect. However, the response to aerosol in a warmer Arctic is different; interactions with aerosols would produce lower LWP clouds, thereby reducing the aerosol cooling effect. Using the Twomey relation (Equation 3.3) and simple assumptions about the future environment, an increase in $N_d$ and move to low LTS conditions implies that the $N_d$-LWP relationship shifts from slightly enhancing the Twomey effect to offsetting it by approximately 50%.

These findings on the dependence of the $N_d$-LWP relationship on the LTS and mean $N_d$ state have important consequences for cloud feedbacks in a warmer, more polluted Arctic. The combined effect of increasing aerosol concentrations in conjunction with increases in surface temperature and decreases in the LTS may ultimately lead to thinner, lower-LWP clouds, with a reduced cooling potential.

*Data availability.* The MODIS data were obtained from NASA Goddard Space Flight Centre (https://modis.gsfc.nasa.gov/data/). The ERA5 data were obtained from the Climate Data Store (https://cds.climate.copernicus.eu/). The sea ice and AMSR-E data were obtained from the National Snow and Ice Data Centre (https://nsidc.org/data/).

*Author contributions.* Both authors contributed to study design and interpretation of results. RMW performed the analysis and prepared the manuscript, with comments from EG.

*Competing interests.* The authors declare that they have no conflict of interest.

*Acknowledgements.* This work was supported by funding from the Royal Society (University Research Fellowship URF/R1/191602) and an Imperial College London Department of Physics PhD studentship.

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
