# Peer review of "Stability dependent increases in liquid water with droplet number in the Arctic"

_Atmospheric Chemistry and Physics, 2021_

## Author Response (AR1)

**Stability dependent increases in liquid water with droplet number in the Arctic**

Rebecca J. Murray-Watson[1] and Edward Gryspeerdt[1]

[1]Space and Atmospheric Physics Group, Imperial College London, UK

**Correspondence:** Rebecca J. Murray-Watson (rebecca.murray-watson17@imperial.ac.uk)

In addition to the edits detailed below in response to the reviewers' comments, we have included a new section on estimating the radiative impact of the clouds in a future Arctic using the Twomey relation; this can be found in Section 3.3 of the new manuscript. We have rewritten our discussion of the 2D histograms, including new analysis on how the precipitation suppression mechanism changes as a function of $r_e$ for a given $N_d$ and LWP. Please note that the line numbers referenced below when discussing where edits have been made correspond to the new manuscript.

**1 Response to Reviewer 1**

**What are the original spatial resolutions for AMSR-E and ERA5 data sets? In addition, it is highly recommended to generate a flow chart or schematic diagram to show how you filtered out cloud data to reduce retrieval bias.**

The text at lines 99 and 150 has been updated to include the original resolutions of the AMSR-E and the ER5 data. A flowchart showing the pixel filtering process has been included in the supplementary information (Figure S1).

**Give the unique environment of Arctic, it is very challenging to obtain accurate cloud information over the Arctic. Therefore, there should be more discussion on uncertainties in satellite-retrieved cloud properties in the Arctic.**

A new section has been added to the methods section with a more thorough discussion of the challenges faced when using satellite retrievals in the Arctic, starting at line 112.

**Figure 1: I would believe that the data samples are highly variable across regions and seasons, given the filtering steps mentioned in Section 2. Can you provide a [separate] spatial map to show the number of samples by season?**

The number of valid retrievals included in each 25 km by 25 km pixel are shown here in Figure 1. When comparing with Figure 1 in the manuscript, it can be seen that while observed the positive sensitivities tend to occur in areas with less data (a product of the fact these locations are often covered by sea ice and experience high solar zenith angles), this is not exclusively the case; there is a positive relationship observed around Svalbard, which also has a high number of points. Additionally, a positive relationship is observed in AMSR-E in areas with more retrievals.

**Section 3.2: There is no explanation on how you separated the positive and negative sensitivity regions. Are there enough and comparable samples in both regions?**

[Figure]

**Figure 1.** Response to reviewer comment, geographic/seasonal distribution of pixels.

The pixels were simply divided into whether they had positive or negative sensitivity based on Figure 1 in the paper. Line 185 has been adjusted to clarify this. In total, there were 5040 pixels with positive sensitivity and 9418 with negative, so we believe that there were enough points to make comparison using the Mann Whitney U test.

**Line 158-162: The explanation on ocean-air temperature gradient and LTS is not clear enough. In general, smaller ocean-air temperature gradient occurs with melting ice in summer, thereby increases the atmospheric stability. In comparison, there is lower atmospheric stability in autumn due to larger ocean-air temperature difference. It is also valid for spring.**

**Reference: A radiation closure study of Arctic stratus cloud microphysical properties using the collocated satellite-surface data and Fu-Liou radiative transfer model: https://agupubs.onlinelibrary.wiley.com/doi/full/10.1002/2016JD025255**

We thank the reviewer for highlighting this reference and have added extra information starting at line 197 for clarity.

**In addition, why do the positive sensitivities occur under high LTS conditions?**

Line 190 has been rephrased for clarity, and additional discussion of the microphysical mechanisms has been included in in the rewritten discussion of Figure 4 at line 234.

**Line 163-170: Have you tried to test this relationship the specific humidity in other vertical levels? Are you expecting any differences?**

Figure 2 shows the relationship (based on Figure 3 in the original manuscript) at 700 hPa and 800 hPa. The dependence on q and LTS are similar to that observed in Figure 3 at 750 hPa, although at 700 hPa, the positive sensitivities stretch to lower values of q.

These figures have been included in the supplementary information and reference to them is now made at line 215.

[Figure]

**Figure 2.** The linear $N_d$-LWP sensitivity plotted as a function of LTS and cloud-top humidity q with q taken at (a) 700 hPa (b) 800 hPa

**Line 191-193: Here, you mentioned that very low $N_d$ conditions are relatively rare in the Arctic and therefore have little impact on the linear sensitivity. Based on $N_d$ histogram in Figure 4, $N_d$ between 50 and 200 (roughly) are more common. However, the LWP-$N_d$ relationship is not clear when $N_d$ falls into [50,200] under high LTS conditions, regardless of high or low q750. It seems to me that there is no any significant LWP-$N_d$ relationship. How do you explain this?**

**Figure 4: Are these relationships different by season?**

The increase in LWP with $N_d$ is difficult to see in the 2D histogram plots due to the scale of the y-axis; we have now included a separate plot showing the average LWP line more clearly (Figure 5 in the new manuscript), for which the increase in LWP with $N_d$ between 50 and 200 cm$^{-3}$ is evident.

Additionally, we have performed the analysis for different seasons; as the 2D histograms are very similar across the seasons, and as such we have only included the average LWP line plots in the manuscript for succinctness. The dependencies on LTS and q hold for the different seasons, as expected from a meteorology-driven effect.

**1.1 Minor comments**

**Line 15: "As the LTS is projected to decrease in a future, warmer Arctic…" Change it to "As the LTS is projected to decrease in a warmer Arctic"**

We have amended the manuscript to your suggestion.

**Line 15: "As the LTS is projected to decrease in a future, warmer Arctic, these results show that aerosol increases may produce lower cloud water paths, offsetting their shortwave cooling effect." If this is the case, what is the overall implication to Arctic climate?**

We have added an additional section (Section 3.3) using the Twomey relation to estimate the radiative impact on the Arctic, finding a potential offset of the Twomey effect of 50%. We have also included extra material at line 330 of the Discussion describing the consequences of a reduced shortwave cooling effect for sea ice loss.

**Line 233-244 and Line 264-266: It would be important to further link the changes in LWP with Arctic sea ice to demonstrate the large-scale impact of liquid cloud on Arctic climate.**

We thank the reviewer for bringing our attention to these papers; the articles on mixed-phase clouds are interesting, and although we focus on liquid clouds here, we plan to look at mixed-phase clouds in future, for which these papers will be relevant. We have added a line to the abstract at line 21 mentioning the implications for sea ice loss and have included extra material at line 330 of the Discussion section describing the consequences in more detail.

**Line 22: "These smaller droplets increase cloud albedo and lead to a shortwave cooling effect." Are you talking about shortwave cooling effect at TOA?**

The text at line 26 has been amended to clarify we are talking about both top of atmosphere and surface cooling.

**Line 43-52: It is better to move this paragraph to the beginning of the Introduction section as it highlights the importance of Arctic clouds and aerosol-induced changes to cloud radiative effects.**

We appreciate the reviewer's suggestion, but feel that the structure of the Introduction as presented, moving from a general discussion about aerosol-cloud interactions to the specifics of the Arctic, outlines our paper well.

**Line 56-57: "Coopman et al. (2016) found that if meteorology is not accounted for, the magnitude of the Arctic clouds response to aerosol is artificially increased by a factor of three." What do you mean by "artificially increased"? Please be specific.**

The text at line 63 has been adjusted for clarity.

**Equation (1): I can't understand that why the breakdown of LWP-AOD relationship by $N_d$ is helpful. As you mentioned that aerosol retrieval is very limited in the Arctic, we will still need $N_d$-AOD ratio to derive LWP/AOD relationship, right?**

Focusing on the $N_d$-LWP component of the LWP-AOD relataionship is helpful as aerosol-induced changes in LWP usually manifest through $N_d$ through mechanisms such as precipitation suppression. Understanding the factors which control the $N_d$-LWP part of the relationship are key to understanding the LWP-aerosol relationship. Also, previous studies (both globally and in the Arctic, such as (Garrett et al., 2004)) have shown that $N_d$ typically increases with aerosol, so the $N_d$-LWP relationship governs the sign of the LWP response to aerosols. Additionally, retrieval of AOD in the Arctic is difficult due to the inability of passive sensors to retrieve AOD and cloud properties in the same pixel.

**Line 132: "This may be due to a potential negative bias in the MODIS data due to retrieval errors (Gryspeerdt et al., 2019)." A negative bias in which variable?**

Line 170 has been changed to make it clear we are discussing a negative bias due to random errors in $r_e$.

**Line 154:" The r2 values of the correlation between the sensitivity is higher" What correlations did you refer to? Please be specific.**

The text at line 192 has been edited for clarity, and an additional figure has been included in the supplementary information (Figure S5) showing the relationships for which the correlation coefficients were calculated. It has also been included below.

[Figure]

**Figure 3.** Scatterplots of the $N_d$-LWP sensitivity for each pixel in Figure 1 (a) plotted against the mean value of (a) LTS (b) $T_{surf}$ and (c) MCAO index for that variable over the 6 years considered. The line of best fit is shown in black and the $r^2$ value is shown above each plot

.

**2 Response to Reviewer 2**

**My primary comment is on the choice of meteorological factors considered in this study. While the authors did a good job of justifying the use of LTS, free tropospheric moisture, and MCAO as meteorological indices by referencing their use in previous studies, I think more discussion of other potential meteorological influences on the LWP-$N_d$ relationship would be beneficial. In the results, LTS is shown to be the most significant of the chosen metrics in predicting whether the LWP-$N_d$ relationship is positive or negative. However, the r2 is still fairly low at 0.39. What other factors, especially those mostly independent of LTS, could be influencing the LWP-$N_d$ relationship, and could there be a better predictor than LTS?**

We appreciate the reviewer's comments, in light of which we have included two extra meteorological variables which are known to influence aerosol-cloud relationships into our analysis; the vertical velocity at 1000 hPa and the surface wind speed, which is in turn related to surface fluxes. The results shown in Figure 2 in the updated manuscript show that these don't seem to strongly influence the relationship, with similar values in the positive and negative regions and low $r^2$ values. The vertical velocity obtained from the reanalysis data would be expected to show a low correlation, especially as it is averaged over a large area.

As to the low value of $r^2$ for LTS (0.39), this may also be explained by other factors which aren't necessarily related to how LTS influences the sensitivity. For example, this may be due to the fact that the LTS is not perfectly represented in the reanalysis data, and if we had more accurate LTS data the $r^2$ may be higher. Additionally, the amount of variance that can be explained in the LWP-$N_d$ relationship depends on errors in the LWP-$N_d$ relationship. Therefore, as we cannot perfectly measure the LWP-$N_d$ sensitivity, the noise in this relationship places an upper bound on the amount of variance that could be explained by any meteorological variable. As such, the LTS explaining 39 % of the variance shows that LTS has a reasonably strong influence on the LWP-$N_d$ relationship.

**Figure 4 shows the most interesting results of this study. I spent quite a bit of time contemplating this figure and I think the discussion of the figure could be improved. First, the authors may want to point out explicitly that LWP begins to decrease with $N_d$ at high $N_d$, high LTS, and low q750. To a lesser extent, the low LTS & low q750 panel shows the same thing as high LTS & low q750, namely, an initial increase in LWP with $N_d$ followed by a decrease in LWP. The big difference is that at high LTS the peak in LWP is around 100/cm3 whereas at low LTS the peak in LWP is at 20/cm3. This difference leads to the interesting patterns in panel h. So, the question to me seems to be why precipitation suppression (which is driving an increase in LWP) at low LTS ends so early. Is it because at low LTS precipitation is weaker for a given $N_d$? Or is it that the drying effects of mixing are much stronger for low LTS and so precipitation suppression is less evident? The latter seems more likely. As written now, there is no discussion of precipitation in explaining panels g and h.**

We thank the reviewer for their insight; we have now rewritten the material discussing Figure 4 and added in an extra plot (Figure 5) to facilitate the interpretation of these results. The new discussion around Figures 4 and 5 (starting at line

234) incorporates the issues raised by the reviewer in their comment; specifically, we have included further discussion of the decreases in LWP for the high LTS/low q panel and extra discussion at line 265 about panels (g) and (h).

To aid with the interpretation of our results, we include extra analysis on how the precipitation suppression mechanism changes as a function of $r_e$ for the aggregated pixels (Figure 6). These data show that in highly polluted conditions, the fraction
135   of precipitating droplets (defined as having $r_e > 15 \ \mu m$) is small, so the precipitation suppression mechanism is weakened. Therefore, entrainment dominates the LWP budget at high $N_d$. The additional mixing with the above-cloud layer in low LTS conditions generates lower LWP clouds, hence explaining the pattern observed in (g) and (h).

**Finally, the authors discuss moisture inversions frequently, but I'm not convinced that they need to be invoked in order to explain anything in this study. For example, moisture inversions are discussed in lines 209-211. But can't the**
140   **clouds in the Arctic have higher LWP at low LTS for the same reasons as discussed in lines 203-206? And generally, I'd think that more moisture above cloud top should reduce evaporation, regardless of whether it is in the form of an inversion or not.**

We thank the reviewer for their points on this matter; we agree that it is unnecessary to invoke moisture inversions to explain the results of this study, and that the results hold simply by considering the absolute humidity as we do elsewhere in the text.
145   As such, the Discussion presented in the new manuscript does not rely on moisture inversions to explain the results.

**2.1   Minor comments**

**Line 12: LTS isn't necessarily the driving force behind spatial variations in LWP response, just the strongest of the metrics studied here. I'm uncomfortable with "driving force" given that the R2 value was somewhat low even for LTS.**

We have amended line 12 as not to overstate the role of LTS in controlling the relationship.

150   **Line 23: Smaller droplets lead to smaller coalescence rates.**

This has been fixed in line 27.

**Line 25: See also Williams and Igel (2021) who argue that smaller droplets radiatively cool cloud top more quickly, generating turbulence, etc. as already stated.**

We thank the reviewer for highlighting this paper and have added in the citation at line 30 of the new manuscript.

155   **Section 2: Can the authors mention somewhere that they're using sunlit times only?**

A sentence at line 136 has been added to reflect this

**Line 130: Caption for Fig 1 states panel (b) is JJA not all seasons. Figure 1: Panel (b) says "AMSR-E all seasons" but caption reads "AMSR-E June, July, and August". Please double check everything for consistency.**

Thank you for pointing out the error-the caption has been corrected.

160   **Line 172: I'm not sure what was meant by this sentence. The "small influence" seems at odds with "a strong response".**

Line 208 has been rewritten to remove the ambiguity.

**Lines 219-220: Not sure what is meant by the background $N_d$ state.**

Upon reflection, we have changed 'background $N_d$ state' to 'mean $N_d$ state' for clarity.

**Section 5: The authors might remind readers that they've only analyzed liquid clouds and not mixed-phase clouds.**

165     We have added in a reminder at line 342 of the new manuscript.

**3 Response to Reviewer 3**

**1) Section 2, Materials: There is no mention about the uncertainty in the retrievals. The authors briefly discussed about it in section 4, but there is no mention about the uncertainty in $\tau_c$ and $r_e$. The errors from this variables can propagate and lead to large uncertainty in LWP and $N_d$ and maybe offset the signals described by the authors. Did the authors looked at this issue? I think it should be properly addressed in the manuscript.**

**2) Section 2, Method: Both $N_d$ and LWP depend only on $r_e$ and $\tau_c$, therefore the two parameters are not independent. I am skeptical about how robust the study is. I think it is robust, but I also think that the two parameters not being independent should be mentioned in the article and the potential issues that might result from that.**

We thank the reviewer for their comments. Further discussion of the uncertainties in $r_e$ and $\tau_c$ have been included in Methods Section; please see our above response to Reviewer 1. Our strict filtering process attempts to reduce the errors associated with $r_e$ and $\tau_c$ by removing cases in which we know the retrievals are uncertain, thereby limiting the impact on LWP and $N_d$. Additionally, as the data have been aggregated into 25 km by 25 km pixels, the effects of random errors in LWP and $N_d$ have been mitigated.

There may be systematic errors that may be generated due to errors in the microphysical property retrievals affecting both $N_d$ and LWP. However, we address this issue by repeating the analysis with AMSR-E, an independent dataset which isn't affected by the correlated errors. The AMSR-E LWP-MODIS $N_d$ sensitivity is shown in Figure 1 of the manuscript, but we have now included analysis of the 2D histograms and the associated average LWP line plots in the supplementary information, also shown below in Figures 4 and 5. The AMSR-E LWP-MODIS $N_d$ relationship shows generally the same trend in meteorology, as the MODIS LWP-$N_d$ relationship. While correlated errors in the MODIS LWP and $N_d$ retrievals might affect the results, the similarity of the results using AMSR-E gives us confidence that this doesn't dominate the relationship.

Additional discussion of the correlated errors in $N_d$ and LWP has been added to the Discussion at line 298.

[Figure]

**Figure 4.** Joint probability histograms for the AMSR-E LWP-MODIS $N_d$ divided into four meteorological regimes based on LTS and q. The difference plots are shown at the end of each row and column, with red over blue indicating higher LWP at higher humidity/LTS. The black lines and grey shading on the joint probability histograms represent the mean LWP value for each $N_d$ bin and the 95 % confidence interval, respectively.

[Figure]

**Figure 5.** The mean ASMR-E LWP for each MODIS $N_d$ bin and the 95 % confidence interval for different meteorological regimes.

**3) l.108, "This stringent filtering is not applied to LWP retrievals". I do not understand how the filter is applied to $N_d$ and not LWP. I thought the colocalization of LWP and $N_d$ was made at the pixel level and one value of LWP requires one value of $N_d$ (especially when retrieving the sensibility). I am wondering if there is something I did not understand in the method. Is it during the spatial resolution averaging (from 1km to 25km)? I am not sure I understand why filtering on one parameters and not the others. Can the author explain ?**

The filtering of the $r_e$ above 4 $\mu$m and $\tau_c$ above 4 happens before the 1 km by 1 km pixels are aggregated to 25 km by 25 km boxes. Therefore, each 25 km by 25 km grid box has $N_d$ retrievals which have been subject to the stricter filtering and LWP retrievals which have not. A flow chart clarifying the filtering process has been included in the supplementary information (Figure S1; see response to Reviewer 1).

Restricting the MODIS LWP to clouds with $\tau_c$ and $r_e$ above 4 would result in a bias towards clouds with higher LWP relative the the AMSR-E LWP. Additionally, as LWP is only linearly dependent on $r_e$, which is the main source of uncertainty, it is less sensitive to uncertainties than $N_d$. Therefore, the LWP is not filtered as strictly as the $N_d$.

**4) l. 92: Is it fair to consider $\gamma$ constant, consider the different conditions in the Arctic? Is $\gamma$ really constant in spring and fall? Can the authors comment on that?**

We thank the reviewer for noting this; we have repeated the analysis using temperature-dependent condensation rate, and found that our previous calculation had underestimated the $N_d$. We have repeated all of our analysis using this new $N_d$ value, but find that it does not significantly affect the sensitivities calculated nor the conclusions of our paper. The Methods section has been updated to reflect our new calculation of $N_d$, and the figures have been updated appropriately.

**5) Section 3: There is no mention on the number of points that the analysis is based on, it is an important information to make sure that the study is statistically robust. I advice the authors to add a plot or supporting information to show that there is enough points, especially to retrieve the sensitivities. For example, in Figure 3, I am skeptical that there is enough points per bin to be robust. Also Figure 3 would benefit to have the 95% confidence interval on the calculation of the sensitivity, or at least a discussion about it.**

We have included a plot showing the number of points for Figure 1 in response to Reviewer 1's comments. As for Figure 3, we omitted in the text that we only included pixels for which there were at least 100 points; we have updated the caption to reflect this. Below we include another plot showing the number of retrievals for each of the q and LTS bins, which has also been added to the supplementary information (Figure S4).

We have also amended Figure 3 to include a measure of significance for the regression; the black dots indicate points for which the correlation is significant at a 95% confidence level.

**3.1  Minor comments**

**1) Title: I find the title of the article confusing and not clear, even after reading the article I am still not convinced by it. A change in the title would benefit to make it more direct.**

We thank the reviewer for their comment, but based on our results and the updated discussion section, we prefer to keep the title as it is.

[Figure]

**Figure 6.** The number of points used for the $N_d$-LWP sensitivity plotted as a function of LTS and cloud-top humidity ($q_{750}$) for Figure (3) in the main manuscript. Only bins which included over 100 valid retrievals were included in the analysis.

**2) There is a lack of quantification in the study, the authors often refer to trend when describing the plots. Also, quantification is needed in the abstract and the conclusion to make the study more robust and appealing for readers.**

**3) For example, the authors described the different variations to be "stronger" (l.190, l.198), "more pronounced" (l. 195)...". Quantification is needed here to support the text and observations.**

We thank the reviewer for their comment; we have rewritten the Discussion section with more quantification in mind. Additionally, we have included a calculation to estimate the effects of the LWP adjustments relative to the Twomey effect to quantify the impact of these findings, which can be found at line 275. Using the 2D joint histograms and the $N_d$ distributions to estimate the present-day LWP distribution, an estimate for $\frac{d\ ln\ LWP}{d\ ln\ Nd}$ can be made by assuming a 10 % increase in the mean $N_d$, representing a hypothetical increase in a polluted Arctic, and generating a new LWP distribution. This analysis was performed for high and low LTS conditions, ignoring the effects of humidity. Using the Twomey relation, the LWP adjustments enhanced the Twomey effect by 9 % in stable conditions, but offset it by 50 % in low LTS environments.

**4) l.22, Twomey, 1977: The publication from Twomey is about the change in albedo rather than the change in cloud droplet size. Therefore the publication is more suited for the next sentence.**

We have fixed the reference at line 26.

**5) l.26: The authors do not mention the effect described by Stevens & Feingold (2009), about the delayed and the offset of the precipitation. Is it because it does not necessarily apply to Arctic clouds? I would mention it in any case.**

**Stevens B, Feingold G. Untangling aerosol effects on clouds and precipitation in a buffered system. Nature. 2009 Oct 1;461(7264):607-13. doi: 10.1038/nature08281. PMID: 19794487.**

We thank the reviewer for bringing this paper to our attention and have made reference to it at line 32.

**6) l.42 "The warming effect of clouds have been linked sea ice loss...": A "with" is missing "been linked with sea ice loss".**

We thank the reviewer for highlighting this, the text at line 49 has been corrected.

**7) l.79: MODIS has issue in the Arctic and often underestimate the cloud top temperature (see Fig. 1 from Tietze et al. (2011)). Can the authors comment on that?**

**Tietze, K., Riedi, J., Stohl, A., and Garrett, T. J.: Space-based evaluation of interactions between aerosols and low-level Arctic clouds during the Spring and Summer of 2008, Atmos. Chem. Phys., 11, 3359–3373, https://doi.org/10.5194/acp-11-3359-2011, 2011.**

Thank you for bringing our attention to this paper; if MODIS is underestimating the cloud top temperature, then it is likely that our lower bound for filtering out liquid clouds is probably removing too many liquid clouds. However, as mixed-phase clouds could introduce significant biases to our results, we will continue to use MODIS's conservative estimate of cloud top temperature for filtering. An acknowledgement of this point is now included on line 85.

**8) l.107, "$\tau_c$ greater than 4". This condition filters a lot of clouds I guess. Can the authors estimate the number of clouds that is filtered out, to understand how representative is it to Arctic clouds?**

We estimate that this criterion filters out about 20% of clouds. While this is a high proportion, it is worthwhile given the added confidence it gives us in the remaining retrievals.

**9) l.148, "Mann Whintey U test": I am not familiar with this test, can the authors explain why it is suited for this case?**

The Mann Whitney U test is a non-parametric test used to compare two independent groups, and the null hypothesis is that there is no difference between the groups. It is suited to this case as the data in two independent categories and are not necessarily normally distributed (especially for the $N_d$ and LWP distributions), although they have a similar shape. It is often described as the non-parametric form of the t-test, and therefore useful for non-normally distributed data. A citation has been included below and at line 187.

H. B. Mann. D. R. Whitney. "On a Test of Whether one of Two Random Variables is Stochastically Larger than the Other." Ann. Math. Statist. 18 (1) 50 - 60, March, 1947. https://doi.org/10.1214/aoms/1177730491

**10) l. 154: I am confused about the $r^2$ parameter, does it refer to the sensitivity of LWP with $N_d$ or LWP with LTS? If the latter, do the authors expect LWP to be linear with LTS, this is not necessarily the case. Same comment about Tsurface and MCAO. Also, I do not understand how mean LTS is taken into account here.**

We have clarified the description at line 192 around the $r^2$ parameter in response to Reviewer 1's comments, and the additional scatterplots shown in the supplementary information. We have updated the manuscript at line 192 for clarity surrounding

use of the mean LTS. The $r^2$ values represent the correlation between the sensitivity and the mean of the LTS and other meteorological variables for each pixel over the six years considered in this study.

**11) l.156, "LTS is used...": Is LTS also correlated with $T_{surf}$ and MCAO? If not, the two other parameters have**

275 **strong sensitivities and might be considered? I am wondering if the authors did the analysis considering different bins of $T_{surf}$ and MCAO.**

LTS is strongly correlated with $T_{surf}$ and MCAO, with Pearson's R of -0.85 and -0.74 respectively. Therefore, repeating the analysis using $T_{surf}$ and MCAO would yield similar results. LTS was chosen due to its higher $r^2$ value.

**12) Figure 2: The description of $r^2$ is missing in the caption?**

280 We have updated our caption of Figure 2 to include a description of $r^2$, in addition to the further description in the text at line 192.

**13) l.176, "assumed a linear sensitivity of LWP to $N_d$": The authors are considering logarithms on Equation 1, so why do they mention it is linear?**

The phrase 'linear sensitivity' is in reference to the fact that this is a linear regression performed in log-log space.

285 **14) l.185, "specific humidity bins": $q_{850}$ has a low correlation and does not seem to be the most important parameter from Figure 2. I am wondering if the authors looked at Figure 3 and 4, considering $T_{surf}$ or MCAO instead of $q_{850}$.**

$T_{surf}$ and MCAO would likely show a similar relationship too LTS, considering their strong correlation (Pearson's R of -0.85 and -0.74) and that in some way they are all proxies for the thermodynamic stability of the boundary layer. Previous work has shown how the role of $q_{750}$ varies for different boundary layer stability conditions (e.g., Figure 1 of Chen et al., 2014),

290 hence the investigation in this study.

**15) l.199, "Figure 4 (f) shows that the difference between...": I do not see it in the Figure. Can the authors rephrase and explicit, (again, maybe quantification would help) ?**

We have rewritten the text for clarity from lines 258 to line 264 to convey our meaning.

**16) Section 4: There is already a lot of discussion in section 3, I would suggest to either move all the discussion from**

295 **section 3 to section 4, or have one section "Results and Discussion".**

While we appreciate the reviewer's suggestion, we find having one section interpreting the results of the analysis and another discussing the wider implications and the potential issues encountered serves the readability of the paper better.

**17) l.358, "Reassessing the Effect of Cloud Type on Earth?s Energy": There is a "?" in the title, change to "Reassessing the Effect of Cloud Type on Earth's Energy".**

300 We thank the reviewer for their attention to detail and we have corrected the citation.

**References**

Chen, Y.-C., Christensen, M., Stephens, G., and Seinfeld, J.: Satellite-based estimate of global aerosol–cloud radiative forcing by marine warm clouds, Nature Geoscience, 7, 643–646, https://doi.org/10.1038/ngeo2214, 2014.

Garrett, T. J., Zhao, C., Dong, X., Mace, G. G., and Hobbs, P. V.: Effects of varying aerosol regimes on low-level Arctic stratus, Geophysical Research Letters, 31, https://doi.org/10.1029/2004GL019928, 2004.

305

---

## Author Response (AR2)

**Response to minor comments: Stability dependent increases in liquid water with droplet number in the Arctic**

Rebecca J. Murray-Watson[1] and Edward Gryspeerdt[1]

[1]Space and Atmospheric Physics Group, Imperial College London, UK

**Correspondence:** Rebecca J. Murray-Watson (rebecca.murray-watson17@imperial.ac.uk)

In addition to the edits detailed below in response to the reviewers' comments, we have fixed two typos in the original manuscript which stated that the enhancement and offset in high and low LTS conditions were calculated as 9% and 51% respectively, when they were in fact 8% and 43% (the values of the slopes quoted were correct). Please note that the line numbers referenced below when discussing where edits have been made correspond to the new manuscript.

**1   Response to Report 1**

**1. Although they have been answered in the Response, it would be nice to include following discussion in the manuscript: 1) why the breakdown of LWP-AOD relationship is important; 2) LWP-Nd relationships and their dependencies on q and LTS are valid in different seasons; 3) Nd-LWP sensitivity as a function of LTS and cloud-top humidity q is valid for q taken at different vertical levels.**

1) The text surrounding the decomposition into the $N_d$-LWP and $N_d$-AOD components around line 70 has been rewritten to include some of the discussion included in the response to reviewers.

2) We have included mention of the robustness of the relationship in different seasons on line 238 of the new manuscript.

3) Reference to analysis done at different vertical levels is made at line 221.

**2. Line 334: "The effects of these LWP changes to the longwave effect, which dominates in non-summer months, is expected to be weaker as changes in longwave downwelling radiation are more strongly controlled by cloud fraction as Arctic clouds in non-summer months typically have LWP greater than 30 g m-2 and therefore act as black-body radiators (Shupe and Intrieri, 2004; Huang et al., 2017, 2019)." This sentence is too long and hard to follow. Please considering rephrase it.**

We thank the reviewer for the suggestion, and the sentence has been rephrased at line 347 of the new manuscript.

**2 Response to Report 2**

**Minor comment: Section 2: On the answer to the reviews, the authors mentioned the uncertainty (on response to reviewer 1), 2)), and I acknowledge the clarifications. Unfortunately, I was hoping for a quantification on the remaining uncertainties: a plot or a range on the uncertainties on tau and re, or retrieved uncertainties on LWP and Nd. I am sure that the different methods decreased the uncertainty but I am still wondering what the remaining uncertainties are.**

We thank the reviewer for their comments, but unfortunately a full understanding of the correlated errors in $N_d$ and LWP generated by uncertainties in $r_e$ and $\tau_c$ are beyond the scope of this work. As we don't have a good idea of how uncertain $r_e$ and $\tau_c$ are in the Arctic, we are unable to analyse the impact of the correlated errors on $N_d$ and LWP. However, in an attempt to understand how these uncertainties may impact our estimate of the offset or enhancement of the Twomey effect, we have used the AMSR-E 2D histograms from the supplementary information and applied the method described in section 3.3. AMSR-E removes the correlated errors between $N_d$ and LWP and therefore offers some constraints (although it has its own errors and uncertainties). In high LTS conditions, the AMSR-E LWP adjustments imply a 13% enhancement of the Twomey effect (slope of +0.05), and under low LTS conditions, an offset of 10% (slope of -0.04). This has been included at line 311 of the new manuscript.

**Technical corrections:**

– l.36: "whereas satellite-based satellite studies" -> "whereas satellite-based studies"

– l. 129: "Dong et al. (2016) saw also saw" -> "Dong et al. (2016) saw also"

– l.216: "(Coopman et al., 2016) found" -> "Coopman et al. (2016) found"

We thank you for pointing out these errors, they have been corrected in the new manuscript.

**Supplementary figures: I find great the addition of the supplementary figures but some of them are not referenced in the main article (Figure S2 and S4). It would be beneficial to reference them in the main article because they support the author's statistics. Supplementary figures: Figure S5 is referenced before Figure S3 in the main article.**

The order of the figures in the supplementary information has been changed so they now appear in the order they are referenced in the text. Figure S2 is now mentioned at line 175, and Figure S4 is now referenced at line 213.

**3 Response to Report 3**

**I have no further comments on the paper. I only note minor grammatical issues in the Figure 6 caption and line 357.**

We thank the reviewer for their attention to detail and we have changed the caption of Figure 6 for clarity.